# Refractive Effect of Epithelial Remodelling in Myopia after Transepithelial Photorefractive Keratectomy

**Diego de Ortueta** [1,*,†] , **Dennis von Rüden** [1] **and Samuel Arba-Mosquera** [2]

1   Aurelios Augenzentrum, 45657 Recklinghausen, Germany
2   SCHWIND Eye-Tech-Solutions, 63801 Kleinostheim, Germany
*   Correspondence: diego.de.ortueta@augenzentrum.org; Tel.: +49-236-1306-9770; Fax: +49-236-1306-9799
†   Current Address: Aurelios Augenlaserzentrum, Erlbruch 34-36, 45657 Recklinghausen, Germany.

**Abstract:** (1) Introduction: We analysed epithelial changes after the treatment of moderate myopia with transepithelial photorefractive keratectomy. (2) Materials and Methods: We used optical coherence tomography data and analysed changes in the stroma and epithelium after ablation. We aimed to ascertain how much epithelium hyperplasia occurred after TransPRK; for this, we used data from 50 eyes treated with TransPRK with the AMARIS 1050 Hz, with a minimum follow-up of 4 months. (3) Results: The measured epithelial changes corresponded to a less than $0.1 \pm 0.2$D of spherical effect, less than $0.2 \pm 0.2$D of astigmatic effect, and less than $0.5 \pm 0.2$D of comatic effect. (4) Conclusions: The changes in epithelial thickness after aberration-neutral transepithelial photorefractive keratectomy for moderate myopia were very small, indicating a low level of epithelial hyperplasia without resembling a regression-inducing lentoid.

**Keywords:** TransPRK; remodelling; epithelial hyperplasia; myopia; photorefractive keratectomy; MS-39; keratometry; regression; aberration-neutral





## 1. Introduction

Since the introduction of laser vision correction (LVC) [1] in the form of either photorefractive keratectomy (PRK) [2], laser-assisted keratomileusis (LASIK) [3], or small-incision lenticule extraction (SMILE) [4], the stability and persistence over time of the resulting correction has been a matter of debate [5]. The evolution of the postoperative refraction back to the initial refraction has usually been termed as refractive regression [6]. However, there have been attempts to distinguish topographic or corneal regression (i.e., the modified corneal curvature changing back to its preoperative curvature) from refractive progression (i.e., the refraction changing back to its preoperative values via, e.g., continued axial elongation) [7].

Topographic or corneal regression has often been associated with changes in the morphology of the epithelium induced by the change in curvature imposed on the stroma [8,9].

It seems that surface ablations such as PRK [10], intrastromal ablations such as LASIK [11], and lenticle extractions such as SMILE [12] are all affected (to different extents) by this induced epithelial hyperplasia, leading to true topographic or corneal regression. Studies of the refractive effects of epithelial remodelling after aberration-neutral transepithelial photorefractive keratectomy (TransPRK) are scarce in the literature [13].

This work aimed to determine the refractive effects of epithelial remodelling after the aberration-neutral transepithelial photorefractive keratectomy (TransPRK) treatment of myopic eyes, beyond the value of the central epithelial thickness.

## 2. Materials and Methods

The records and charts of 50 eyes of 27 patients treated with an aberration-neutral [14] TransPRK [15] profile for the correction of myopia with or without astigmatism were evaluated.

The evaluation involved a comparison of the preoperative data with the data at 4 months. We analysed the standard refractive outcomes reported in LVC [16], along with an analysis of the changes in corneal curvature at 4 months (for 3, 5, and 7 mm diameters, in order to also determine local deviations from the achieved refractive correction) [17], and an evaluation of the changes in the central corneal thickness (CCT), changes in the central stromal thickness (CST), and changes in the epithelial thickness for the horizontal and vertical meridians at 6 and 8 mm diameters (nine values) [18].

The corneal topographic and tomographic scans were obtained using an MS-39 (Costruzione Strumenti Oftalmici, Firenze, Italy) optical coherence tomography (OCT) device [19]. For the described calculations, we used the refractive outcomes as well as the topographic findings of the anterior surface topography and stroma topography to measure the change in keratometric power, using a method described in a previous paper [20].

Visual acuities (UDVA and CDVA) were measured using Snellen charts following the ISO 10938:2016 standard [21], converted into the logarithmic of the minimum angle of resolutions (logMAR) for analysis, and reported in 20/x Snellen equivalent fractions for better comparability. Refractions are expressed in the form of spherical equivalent (SEq) and cylinder. Corneal curvature was evaluated in keratometric dioptres using the (flat and steep) meridian values provided by the CSO MS-39 for simulated keratometries as well as the 3, 5, and 7 mm diameters. Changes in corneal epithelial values were expressed in terms of change in axial dioptres (due to changes in epithelial thickness).

We calculated the refractive power associated with the change in epithelial thickness based on a simplified parabolic approximation of the Munnerlyn equation. This simple refractive model resulted in the following equation:

$$Hemimeridional\,Power(r) = \frac{2 * \left(n_{Epi} - n_{Air}\right)}{r^2}\left(\Delta Epi_{post-pre}(r) - \Delta Epi_{post-pre}(0)\right)$$

where $n$ is the refractive indices of epithelium and air. For our calculations, we considered $n$Cornea = 1.401 and $n$Air = 1.

In order to provide a better degree of information, it is also feasible to depict the wavefront aberration in optical refractive power without having to reduce it to a quadric surface. A simple solution to the issue is to apply the idea of axial refractive error (i.e., vergence maps). The wavefront aberration is zero at the pupil centre and perpendicular to the line of sight, which represents a main ray. The wavefront's local surface is perpendicular to each point's direction of propagation. The axial refractive error is expressed as the distance in dioptres from the centre of the pupil to the point where the propagating local wavefront and the line of sight intersect.

The SEq of the epi remodelling can be estimated as the average of the axial power of all measured points. The astigmatism component of the epi remodelling can be estimated as the difference of the average of the axial power along the horizontal and vertical meridians. The horizontal and vertical coma components of the epi remodelling can be estimated as the difference of the average of the axial power along opposed hemimeridians.

All of the treatments were performed between September 2020 and Mai 2021 by a single surgeon, DdO, using non-wavefront-guided but aspheric aberration-neutral ablation profiles in TransPRK, using SCHWIND AMARIS 1050RS [22]. The surgical technique has been previously described [22].

There were no specific inclusion or exclusion criteria for the retrospective chart review other than having received TransPRK treatment for myopia until −6 D and having a completed 4-month follow-up. The charts were selected as consecutive cases over the mentioned period. Some cases had just one eye included; this may be due to the patient having received a unilateral procedure (with the contralateral eye having the minimum refraction or actual emmetropia) or a nonmyopic procedure in the contralateral eye (mixed astigmatism, simple or compound hyperopic astigmatism, or simple hyperopia). The demographic data are presented in Table 1.

**Table 1.** Demographic data.

| Parameter (Unit) | Preoperative Value | Postoperative Value | p-Value |
|---|---|---|---|
| Number of eyes | 50 | 50 | - - - |
| Laterality OD (%) | 48% | 48% | - - - |
| UDVA (logMAR) | 0.9 ± 0.5 (0.2 to 2.0) | 0.0 ± 0.1 (−0.1 to 0.1) | <0.0001 |
| Manifest sphere (D) | −2.41 ± 1.49 −6.00 to 0.00) | 0.18 ± 0.23 (−0.25 to 0.75) | <0.0001 |
| Manifest cylinder (D) | −0.80 ± 0.69 (−2.50 to 0.00) | −0.11 ± 0.19 (−0.75 to 0.00) | <0.0001 |
| CDVA (logMAR) | 0.0 ± 0.1 (−0.1 to 0.2) | 0.0 ± 0.1 (−0.1 to 0.1) | 0.2 |
| Target sphere (D) | 0.05 ± 0.11 (−0.10 to 0.32) | - - - | - - - |
| Optical zone (mm) | 6.9 ± 0.2 (6.5 to 7.5) | - - - | - - - |
| Ablation depth (μm) | 111 ± 23 (75 to 162) | - - - | - - - |
| 3 mm flat keratometry (D) | 42.9 ± 1.7 (39.6 to 47.2) | 40.2 ± 2.4 35.6 to 46.4) | <0.0001 |
| 3 mm steep keratometry (D) | 43.8 ± 1.9 (41.1 to 49.7) | 41.0 ± 2.5 (36.0 to 47.0) | <0.0001 |
| Thinnest epithelium (μm) | 47 ± 3 (42 to 53) | 48 ± 4 (39 to 54) | 0.006 |
| X-position thinnest epithelium (μm) | 46 ± 209 (−272 to 280) | 27 ± 176 (−277 to 207) | 0.2 |
| Y-position thinnest epithelium (μm) | 190 ± 123 (−216 to 280) | 32 ± 234 (−280 to 279) | 0.008 |
| Central epithelium thickness (μm) | 54 ± 3 (48 to 58) | 56 ± 3 (47 to 61) | 0.0002 |
| 3 mm nasal epithelium thickness (μm) | 53 ± 3 (47 to 59) | 55 ± 3 (44 to 61) | 0.0001 |
| 3 mm temporal epithelium thickness (μm) | 52 ± 3 (46 to 59) | 56 ± 3 (46 to 63) | <0.0001 |
| 3 mm superior epithelium thickness (μm) | 51 ± 3 (46 to 57) | 54 ± 4 (44 to 61) | 0.0001 |
| 3 mm inferior epithelium thickness (μm) | 53 ± 3 (47 to 60) | 54 ± 3 (44 to 60) | 0.02 |
| 4 mm nasal epithelium thickness (μm) | 51 ± 3 (45 to 56) | 52 ± 3 (46 to 58) | 0.0007 |
| 4 mm temporal epithelium thickness (μm) | 50 ± 3 (44 to 57) | 52 ± 3 (46 to 58) | <0.0001 |
| 4 mm superior epithelium thickness (μm) | 46 ± 3 (40 to 55) | 49 ± 4 (38 to 56) | <0.0001 |
| 4 mm inferior epithelium thickness (μm) | 51 ± 3 (46 to 58) | 52 ± 3 (42 to 58) | 0.4 |
| Central corneal thickness (μm) | 551 ± 30 (490 to 628) | 484 ± 42 (405 to 582) | <0.0001 |

Refractive and Keratometry data preoperatively and at 4 months postoperatively after TransPRK treatment.

We considered both eyes per patient (when available) to increase the sample size. In order to account for this, the statistical analyses used the number of patients instead of the number of eyes to determine the significance, but retained the data on all eyes to determine the mean values. Paired Student *t*-tests were performed to compare the postoperative to preoperative changes in the parameters.

### 3. Results

The records and charts of 50 eyes of 27 patients treated consecutively within an eight-month period with an aberration-neutral TransPRK profile for the correction of myopia with or without astigmatism were evaluated at the 4-month follow-up.

The patient and treatment demographics are presented in Table 1.

The standard refractive outcomes reported in LVC are presented in Figure 1A–H. In summary, 100% of the eyes reached 20/25 or better postoperatively (Figure 1A) and 98% of the eyes had a postoperative UDVA within one line of the preoperative CDVA (Figure 1B); no single eye lost two or more lines of CDVA postoperatively (Figure 1C); the correction of the SEq was slightly overcorrected by ~+4% (Figure 1D), with 96% of the eyes having a postoperative SEq within 0.5D of the intended target (Figure 1E) and 98% of the eyes having a postoperative refractive cylinder of 0.5D or less (Figure 1F); and the correction of the refractive cylinder was slightly overcorrected by ~+9% (Figure 1G), with all eyes within 15 deg of the target induced astigmatism axis (Figure 1H).

Changes in corneal keratometries according to treatment plan are presented in Figure 2A–D. In summary, the refractive overcorrection of the SEq was retrieved for the 3 mm diameter corneal keratometries and the correction slightly reduced for larger diameters (5 and 7 mm) (Figure 2A), with over 78% of the eyes having postoperative keratometries within 0.5D of the intended target (Figure 2B) and over 74% of the eyes having a postoperative corneal toricity of 1.0D or less (Figure 2C); the refractive overcorrection of the cylinder was retrieved for the 3 mm diameter corneal keratometries and the correction was slightly reduced for larger diameters (5 and 7 mm) (Figure 2D).

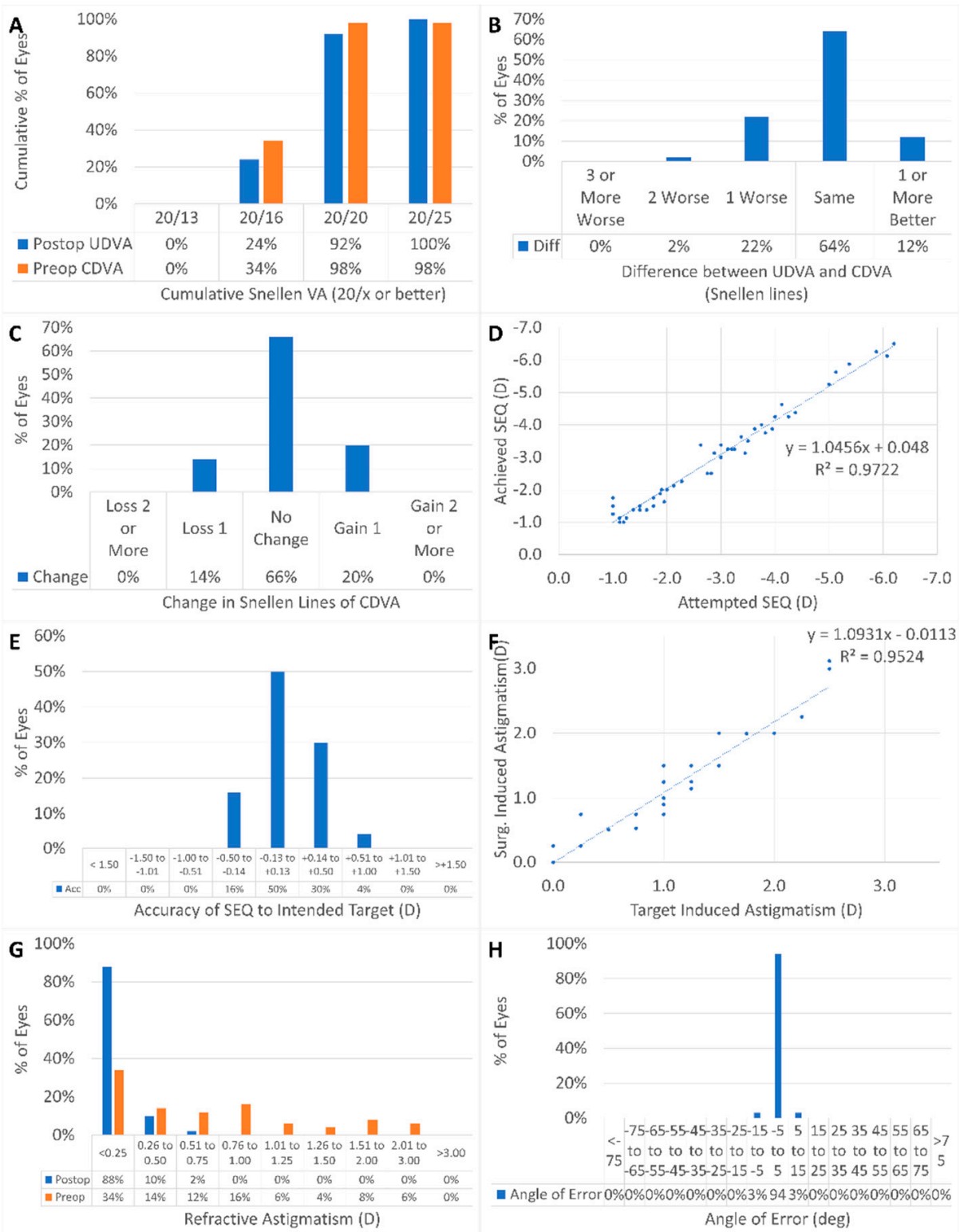

**Figure 1.** 100% of the eyes reached 20/25 or better postoperatively (**A**) and 98% of the eyes had a postoperative UDVA within one line of the preoperative CDVA (**B**); no single eye lost two or more lines of CDVA postoperatively (**C**); the correction of the SEq was slightly overcorrected by ~+4% (**D**), with 96% of the eyes having a postoperative SEq within 0.5D of the intended target (**E**) and 98% of the eyes having a postoperative refractive cylinder of 0.5D or less (**F**); and the correction of the refractive cylinder was slightly overcorrected by ~+9% (**G**), with all eyes within 15 deg of the target induced astigmatism axis (**H**).

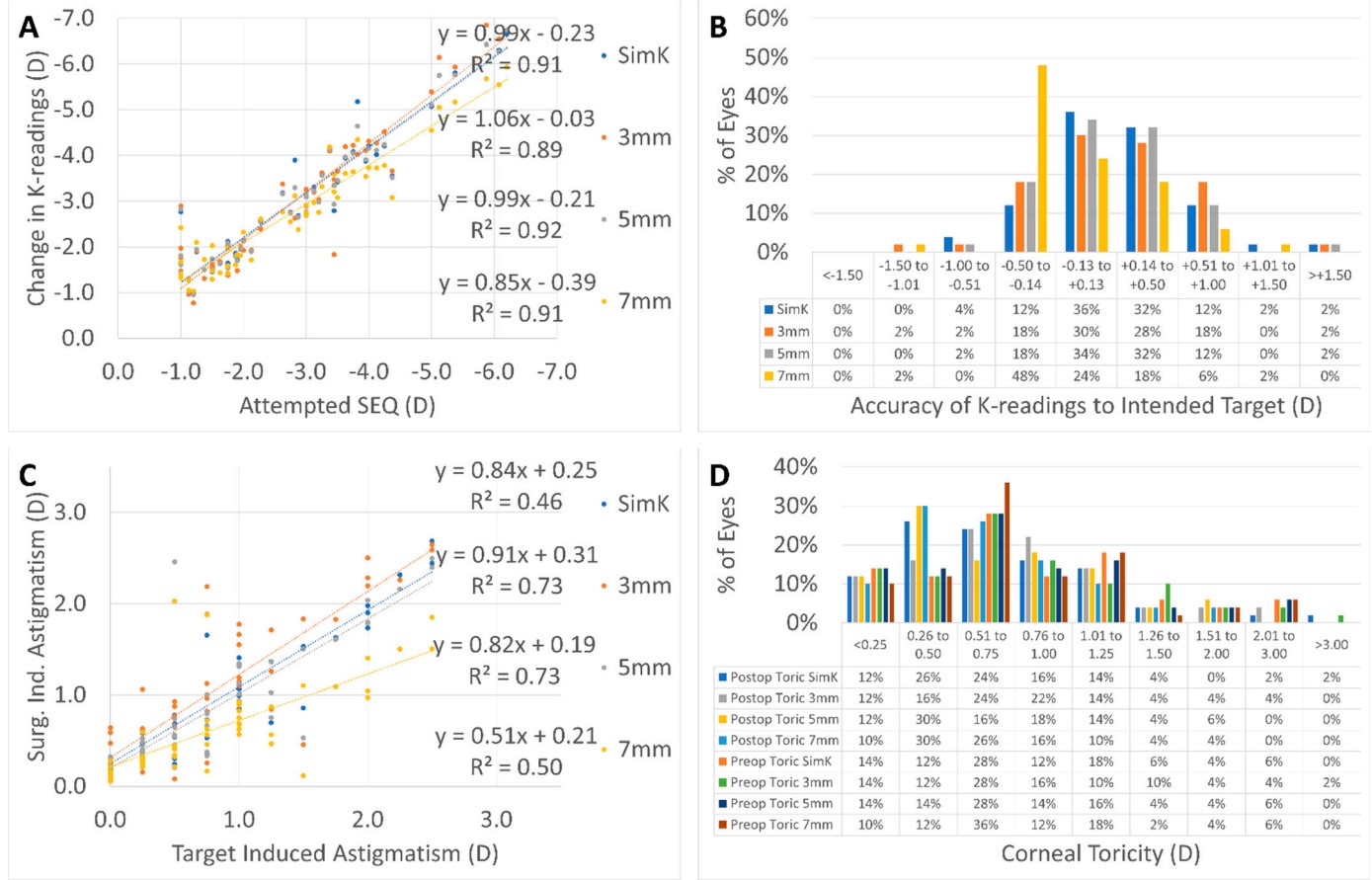

**Figure 2.** The refractive overcorrection of the SEq was retrieved for the 3 mm diameter corneal keratometries and the correction slightly reduced for larger diameters (5 and 7 mm) (**A**), with over 78% of the eyes having postoperative keratometries within 0.5D of the intended target (**B**) and over 74% of the eyes having a postoperative corneal toricity of 1.0D or less (**C**); the refractive overcorrection of the cylinder was retrieved for the 3 mm diameter corneal keratometries and the correction was slightly reduced for larger diameters (5 and 7 mm) (**D**).

Figure 3 shows the changes in the central thickness (total cornea and stromal thickness) in relation to the attempted central stromal ablation (the total central ablation minus the epithelial thickness incorporated into the TransPRK profile). The central corneal thickness was reduced to $6 \pm 11$ µm more than the attempted central stromal ablation, whereas the over ablation reached $8 \pm 10$ µm more for the central stromal thickness.

The preoperative and postoperative epithelial thickness profiles for the horizontal and vertical meridians, along with the differences between them, are depicted in Figure 4 (panels A and B). In summary, the epithelium thickened on average by $2 \pm 4$ µm postoperatively, without clear signs of a lentoid refractive shape (Figure 4A,B).

Finally, the refractive effect of epithelial remodelling after aberration-neutral transepithelial photorefractive keratectomy is presented in Figure 5. The refractive effect of epithelial remodelling after aberration-neutral transepithelial photorefractive keratectomy remained within 0.25D for all measured locations in both meridians. Combining the effects of both meridians, a 0.09D spherical equivalent component (average of the horizontal and vertical meridional effects,) together with a 0.19D astigmatism component (difference between the horizontal and vertical meridional effects) and 0.49D coma component (peak to valley refractive difference along the meridians), could be estimated as the overall refractive effect of epithelial remodelling after aberration-neutral transepithelial photorefractive keratectomy.

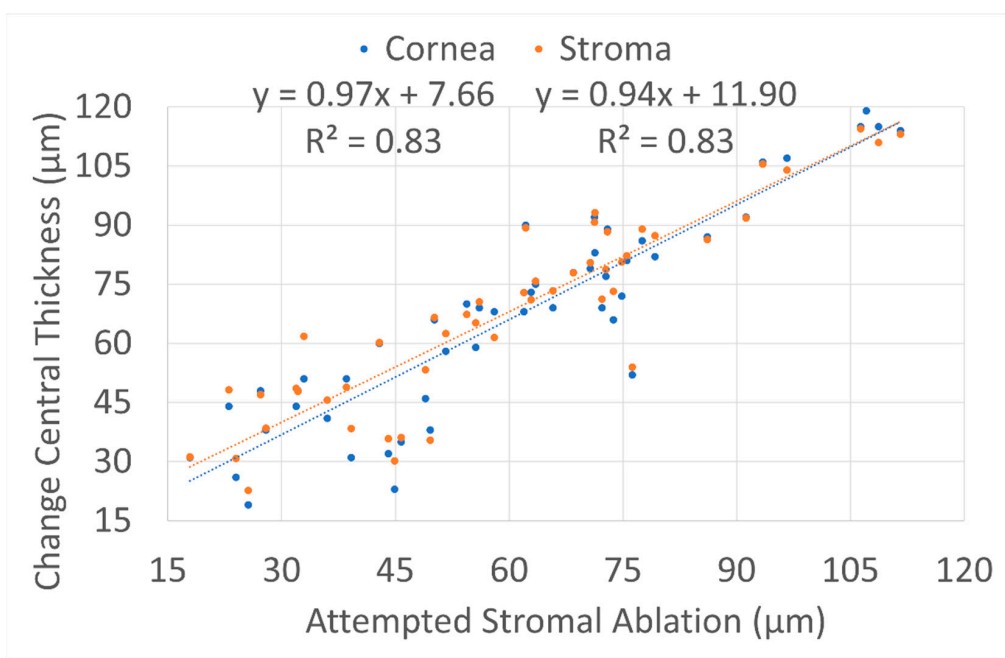

**Figure 3.** Central thickness (total cornea and stromal thickness) in relation to the attempted central stromal ablation (the total central ablation minus the epithelial thickness incorporated into the TransPRK profile). The central corneal thickness was reduced to $6 \pm 11$ μm more than the attempted central stromal ablation, whereas the over ablation reached $8 \pm 10$ μm more for the central stromal thickness.

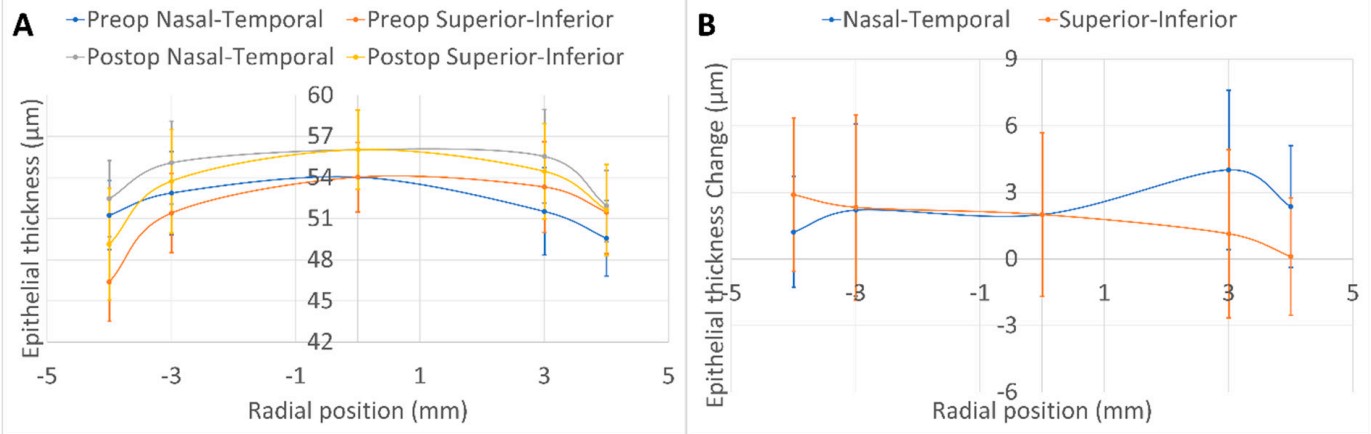

**Figure 4.** The epithelium thickened on average by $2 \pm 4$ μm postoperatively, without clear signs of a lentoid refractive shape.

There was no statistically significant correlation between the amount of correction (considered here as refractive power and stromal ablation depth, but also maximum peripheral slope) and the induced hyperplasia (considered in μm as well as its associated refractive effect).

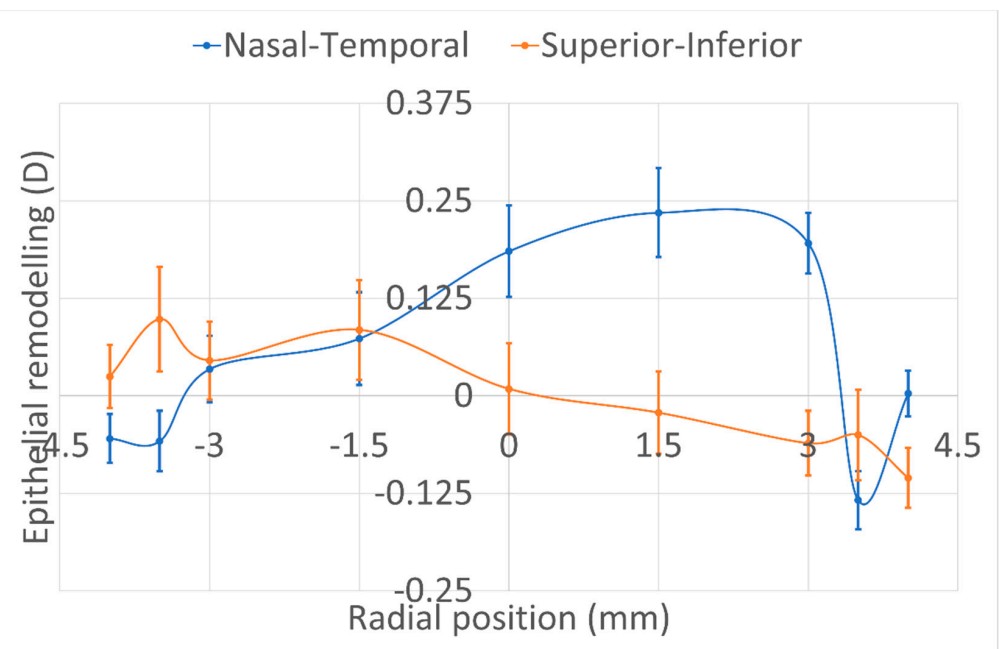

**Figure 5.** The refractive effect of epithelial remodelling after aberration-neutral transepithelial photorefractive keratectomy remained within 0.25D for all measured locations in both meridians. Combining the effects of both meridians, a 0.09D spherical equivalent component (average of the horizontal and vertical meridional effects,) together with a 0.19D astigmatism component (difference between the horizontal and vertical meridional effects) and 0.49D coma component (peak to valley refractive difference along the meridians), could be estimated as the overall refractive effect of epithelial remodelling after aberration-neutral transepithelial photorefractive keratectomy.

## 4. Discussion

The short-term stability of LVC has been questioned in the past [23]. In particular, PRK seems to be associated with an overshoot followed by a transient reduction in the overcorrection toward a "final" refraction [24]. This has been considered as "part of the technique", but is simultaneously a weak point of PRK compared to LASIK or lenticule extraction (SMILE, SmartSight) in which the "final" refraction is obtained much faster [25]. The same holds for hyperopic vs. myopic corrections, hyperopic PRK being the treatment with the longest recovery time.

Previous studies on both PRK and LASIK have aimed to distinguish topographic or corneal regression from refractive progression [26]. We used refractive outcomes as well as topographic findings to measure the change in keratometric power, according to a method described in a previous paper [20], as we were able to measure the epithelial thickness and stroma thickness and obtain both curvatures with the OCT system. The advantage of using keratometric data is that they offer an objective method with which to calculate changes to the cornea.

The epithelium seems the most logical candidate to explain short-term to mid-term changes at the anterior corneal surface [10]. Epithelial hyperplasia occurs for all kinds of treatments, PRK [10], LASIK [27], and SMILE [28].

This study evaluated 50 eyes of 27 patients consecutively treated with an aberration-neutral TransPRK profile for the correction of myopia until −6D with or without astigmatism, namely the spatially resolved epithelial thickness before and after Trans-PRK. In the statistical analysis, we used the number of patients instead of the number of eyes to determine the significance, but retained data on all eyes to determine the mean values.

Outcomes were reported at the 4-month follow-up, as previous longitudinal studies showed no further changes in epithelial remodelling past the 1-month follow-up (i.e., no difference between 1-month and 12-month follow-ups) [29].

The evaluation involved the standard refractive outcomes reported in LVC. In our group of treated myopic eyes, 100% reached 20/25 or better postoperatively and no single eye lost two or more lines of CDVA postoperatively. Furthermore, 96% of the eyes had a postoperative SEq within 0.5D of the intended target and 98% of the eyes had a postoperative refractive cylinder of 0.5D or less.

The changes in corneal curvature showed that refractive correction was properly obtained within the planned optical zone (OZ) and only reduced for larger diameters (7 mm) outside of the OZ and encompassing the transition zone (TZ) (Figure 2A,D). A low induction of aberrations (spherical aberration and higher-order astigmatism) can be inferred from this analysis.

The changes in CCT and CST were very similar as shown in the example of an eye Figure 6. (reinforcing the finding that low epithelial hyperplasia was induced). Furthermore, these changes were statistically highly compatible with the refractive and keratometric findings, with all three analysis types—refractive, keratometric, and thickness—showing a similar level of minor statistical overcorrection.

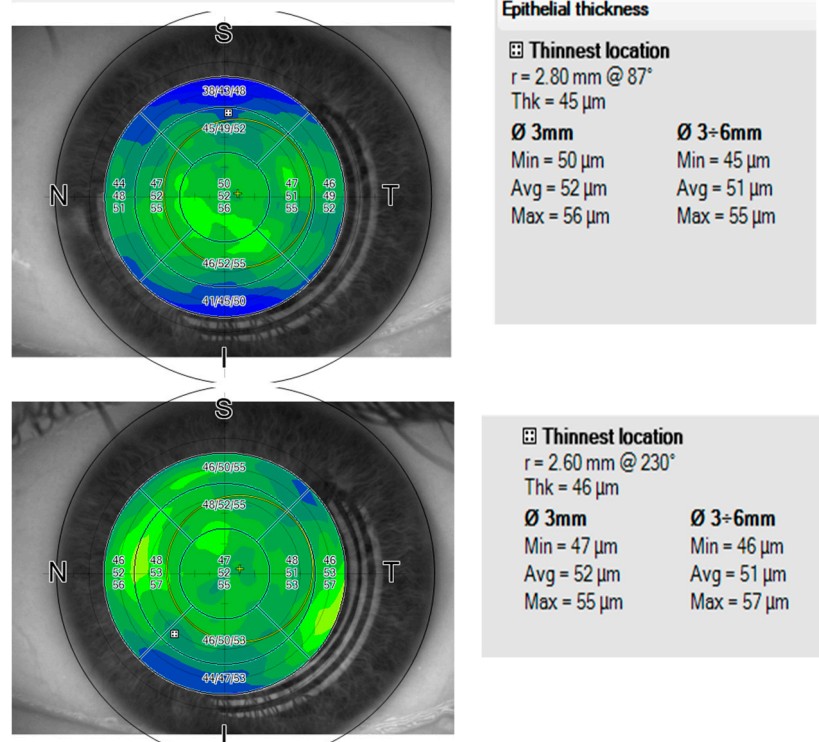

**Figure 6.** The image (top left) showing the preoperative epithelial thickness map OCT preoperatively and (bottom left) postoperatively after 4 months. At the right are the summary epithelial indices.

The changes in epithelial thickness for the horizontal and vertical meridians at 6 and 8 mm diameters (nine values including the centre) were very small, indicating a low level of epithelial hyperplasia (~2 μm). Furthermore, the changes more resembled an overall thickening than a regression-inducing lentoid [30]. When expressing the measured changes in equivalent refractive dioptres, it was possible to determine less than 0.1D of spherical effect, along with less than 0.2D of astigmatic effect, and less than 0.5D of comatic effect.

This low effect may also be associated with the moderate myopic corrections included in this cohort, and with the demonstrated good aspheric behaviour of the studied platform for low and moderate myopic corrections, together with a large optical of at least 6.75 mm and a progressive transition zone of at least 2 mm [31].

A previous longitudinal study of TransPRK using a similar system also demonstrated essentially no epithelial thickening at 1-month follow-up [32]. Another study using the

same platform as the current work, interestingly, found that thinner epithelia tended to thicken more after TransPRK compared to preoperatively thicker epithelia (which tended to thin after TransPRK) [33].

Unlike previous studies, we could not find any statistically significant correlation between the amount of correction (considered in terms of refractive power and stromal ablation depth, but also maximum peripheral slope) and the induced hyperplasia (considered in μm as well as its associated refractive effect) [30].

Due to the inherent spot size and other considerations, surface ablation causes some degree of corneal surface roughness [34–36]. An epithelial response to postoperative irregularity of the corneal surface reduces the overall roughness [37,38]. Following such a surgery, the epithelium will grow back in an attempt to make up for the changes (and created abnormalities) the laser ablation made to the cornea. In a perfect world, epithelial renewal would lessen laser-induced abnormalities, which would lessen the eye's overall corneal aberrations [39].

Epithelial thickness modulations following ablation can be theoretically calculated, according to Huang et al., to explain clinically observed regression and the induction of aberration [40]. We believe that a smooth stromal bed [41], as we obtained with our laser platform after the ablation, together with a large OZ [42] and TZ [43], produces less remodelling of the epithelium.

Growing evidence supports the significance of the corneal epithelium's contribution to total ocular refraction and corneal net power. Epithelial refractive power alone has been reported to be 0.85D (range 0.29–1.60D) in the 3.6 mm diameter zone and 1.03D (average, range 0.55–1.85D) at the central 2 mm diameter zone [44]. Other reports have found lower values for the optical power of the epithelium [45].

It has been suggested in the past that regression after PRK may be caused by remodelling of the stroma rather than by epithelial hyperplasia [46,47]. However, this effect is probably a long-term drift (stromal remodelling) that cannot be determined in the early 4-month follow-up. Epithelial hyperplasia occurs to conceal irregularities of the stromal surface after PRK, and as shown here, seems to play a minimal role, at least after TransPRK using smooth-ablation algorithms. This may be different after LASIK or lenticule extraction procedures. The extent of postoperative epithelial hyperplasia depends on the amount of correction (flattening of the central cornea).

The fact that this was a retrospective study with a limited number of cases with a 4-month follow-up may be considered an inherent limitation. Including other control groups (e.g., using non optimised ablation profiles) would help answer the question of whether the aberration-neutral profile or SmartPulse Technology led to less epithelial remodelling and clarify the corresponding refractive effect.

## 5. Conclusions

In conclusion, the changes in epithelial thickness after aberration-neutral transepithelial photorefractive keratectomy were very small, indicating a low level of epithelial hyperplasia (~2 μm) without resembling a regression-inducing lentoid. When expressing the measured changes in equivalent refractive dioptres, it was possible to determine less than 0.1D of spherical effect, along with less than 0.2D of astigmatic effect, and less than 0.5D of comatic effect.

**Author Contributions:** Conceptualisation, D.d.O. and S.A.-M.; Methodology, D.d.O. and S.A.-M.; Validation, D.d.O. and S.A.-M.; Formal analysis, D.d.O. and D.v.R.; Investigation, D.d.O.; Data curation, D.v.R.; Writing—original draft preparation, D.d.O.; Writing—review and editing, D.d.O. and S.A.-M.; Visualisation, D.d.O., D.v.R. and S.A.-M.; Supervision, D.d.O.; Project administration, D.d.O. All authors have read and agreed to the published version of the manuscript.

**Funding:** This research received no external funding.

**Institutional Review Board Statement:** The study was conducted according to the guidelines of the Declaration of Helsinki. The study was retrospective with anonymised data. Informed consent was obtained from all subjects involved in the study. According to the Medical College ÄKWL § 15 Paragraph 1, retrospective anonymised data do not require ethical approval. (https://www.medizin.uni-muenster.de/ek/ethik-kommission/antragsunterlagen/unterlagen-fuer-antraege/ausnahmen.html). Accessed 29 September 2021.

**Informed Consent Statement:** Informed consent was obtained from all the subjects involved in the study.

**Data Availability Statement:** All the data were fully anonymised and are available upon request.

**Conflicts of Interest:** D.d.O. is a consultant for SCHWIND eye-tech-solutions GmbH, S.A.-M. is an employee of SCHWIND Eye-Tech-Solutions; D.v.R. declares no conflicts of interest.

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
