# Peer review of "Refractive Effect of Epithelial Remodelling in Myopia after Transepithelial Photorefractive Keratectomy"

_2411-5150, 2022_

Round 1

Reviewer 1 Report

It is a interesting topic. well structuret. good results.

Good job!

Author Response

Thank you for the review

Reviewer 2 Report

This paper analyzed the changes of the corneal epithelial thickness and the correlation to the refractive changes after TransPRK, which is similar to Corneal Epithelial Remodeling and Its Effect on Corneal Asphericity after Transepithelial Photorefractive Keratectomy for Myopia published on J Ophthalmol in 2016.

Please see below my comments relating to specific aspects.

Materials and Methods:

1. Some of the selected patients were included in the study with simple eye, should all subjects be studied with simple eye?

2. Lack of the informed consent and ethical approval.

3. Lack of statistical analysis.

Results

1. Adding typical OCT images of corneal epithelium before and after surgery.

2.The author compared the relationship between the depth of ablation and corneal epithelial thickness only, while he did not compared whether there was a correlation between the depth of ablation and the refractive effects.  

References: Remove and replace the outdated documents.

Author Response

Dear Reviewer

Thank you for the comments and idea that you proposed.

We have answered your questions and make the changes you proposed .

  1. Some of the selected patients were included in the study with simple eye, should all subjects be studied with simple eye?

You can find following text in methods

"We considered both eyes per patient (when available) to increase the sample size. In order to account for this, the statistical analyses used the number of patients instead of number of eyes to determine the significance, but retained data on all eyes to determine the mean values."

 And practically presented, e.g. Table 1 showing p values

End of Results:

"There was no statistically significant correlation between the amount of correction (considered here as refractive power and stromal ablation depth, but also maximum pe-ripheral slope) and the induced hyperplasia (considered in µm as well as its associated refractive effect).

  1. Lack of the informed consent and ethical approval.

At the end of the paper we wote:

Ethical approval and consent to participate: The study was conducted according to the guidelines of the Declaration of Helsinki. The study was retrospective with anonymized data. Informed consent was obtained from all subjects involved in the study. According to the Medical College ÄKWL § 15 Paragraph 1 retrospective anonymized data does not require ethical approval. (https://www.medizin.uni-muenster.de/ek/ethik-kommission/antragsunterlagen/unterlagen-fuer-antraege/ausnahmen.html) We addad the link in the paper.

  1. Lack of statistical analysis.

Statistical analysis was formally summarized in the methods section at the end

"We considered both eyes per patient (when available) to increase the sample size. In order to account for this, the statistical analyses used the number of patients instead of number of eyes to determine the significance, but retained data on all eyes to determine the mean values."

And practically presented, e.g. Table 1 showing p values

End of Results:

"There was no statistically significant correlation between the amount of correction (considered here as refractive power and stromal ablation depth, but also maximum pe-ripheral slope) and the induced hyperplasia (considered in µm as well as its associated refractive effect)."

Discussion:

"In the statistical analysis, we considered only one eye per patient to further strengthen the statistical power. In order to account for this, the statistical analyses used the number of patients instead of the number of eyes to determine the significance, but retained data on all eyes to determine the mean values."

"The changes in CCT and CST were very similar (reinforcing the finding that low epithelial hyperplasia was induced). Furthermore, these changes were statistically highly compat-ible with the refractive and keratometric findings, with all three analysis types—refractive, keratometric, and thickness—showing a similar level of minor statis-tical overcorrection."

"Unlike previous studies, we could not find any statistically significant correlation be-tween the amount of correction (considered in terms of refractive power and stromal ablation depth, but also maximum peripheral slope) and the induced hyperplasia (con-sidered in µm as well as its associated refractive effect) [31]."

Methods has been now extended to:

"Paired Student T-tests have been performed to compare postoperative to preoperative changes in the parameters"

  1. Adding typical OCT images of corneal epithelium before and after surgery.

Thank you for this idea, we added an OCT image with Epithelium preoperative and postoperative epithelium thickness.

5 .The author compared the relationship between the depth of ablation and corneal epithelial thickness only, while he did not compared whether there was a correlation between the depth of ablation and the refractive effects. 

We actually did as expressed in Results and Methods.

At the End of Results:

"There was no statistically significant correlation between the amount of correction (considered here as refractive power and stromal ablation depth, but also maximum peripheral slope) and the induced hyperplasia (considered in µm as well as its associated refractive effect)."

And in the Discussion:

"Unlike previous studies, we could not find any statistically significant correlation be-tween the amount of correction (considered in terms of refractive power and stromal ablation depth, but also maximum peripheral slope) and the induced hyperplasia (considered in µm as well as its associated refractive effect) [31]."

Round 2

Reviewer 2 Report

Thank you for submitting your revised manuscript. Although you have done a lot of work, this research is not innovative enough to be published in this journal.